

**African dust transported to Barbados in the Wintertime Lacks Indicators of Chemical**
**Aging**
Haley M. Royer[1,2], Michael Sheridan[1,3], Hope Elliott[1,4], Nurun Nahar Lata[5], Zezhen Cheng[5],
Swarup China[5], Zihua Zhu[5], Andrew P. Ault[6], Cassandra Gaston[1*]
[1]Department of Atmospheric Sciences, Rosenstiel School of Marine, Atmospheric, and Earth
Science, University of Miami, Miami, FL
[2]Department of Environmental Sciences and Engineering, University of North Carolina at
Chapel Hill, Chapel Hill, NC
[3]Skidaway Institute of Oceanography, University of Georgia, Athens, GA
[4]Department of Ocean Sciences, Rosenstiel School of Marine, Atmospheric, and Earth Sciences,
University of Miami, Miami, FL
[5]Environmental Molecular Sciences Laboratory, Pacific Northwest National Laboratory,
Richland, WA
[6]Department of Chemistry, University of Michigan, Ann Arbor, MI
*Corresponding Author:
Cassandra J. Gaston: Email: cgaston@miami.edu, Phone: (305)-421-4979





## 1. Abstract

The chemical processing ("aging") of mineral dust is thought to increase dust light scattering efficiency, cloud droplet activation, and nutrient solubility. However, the extent of African dust aging during long-range transport to the western Atlantic is poorly understood. Here, we explore African dust aging in wintertime samples collected from Barbados when dust is transported at lower altitudes. Ion chromatography (IC) analysis of bulk nitrate, sulfate, and oxalate increase when African dust reaches Barbados, indicating dust aging. However, aerosol mixing state analysis from computer-controlled scanning electron microscopy with energy dispersive x-ray spectroscopy (CCSEM/EDX) indicates that approximately 67% of dust particles are internally mixed with sea salt, while only about 26% of dust particles contain no internally mixed components. SEM/EDX elemental mapping and time-of-flight secondary ion mass spectrometry (TOF-SIMS) reveals that within internally mixed dust and sea salt particles, only sea salt components contain signs of aging.



## 2. Introduction

Upon emission, dust can directly scatter or absorb solar radiation (Balkanski et al., 2007; Haywood et al., 2003; Myhre & Stordal, 2001; Sokolik et al., 2001; Tegen, 2003), act as cloud condensation (Albrecht, 1989; Koehler et al., 2009; Zev Levin et al., 1996; Rosenfeld et al., 2001) and ice nuclei (Archuleta et al., 2005; Cziczo et al., 2004; DeMott et al., 2003), and provide micro- and macro- nutrients to nutrient-limited ecosystems (Elliott et al., 2024; Jickells et al., 2005; Mahowald, 2011). However, mineral dust aerosol is still poorly represented in climate models due, in part, to a lack of understanding of the physiochemical properties of dust and their changes as a result of cloud processing (Gierlus et al., 2012; Wurzler et al., 2000), multiphase reactions (Andreae et al., 1986; Sullivan et al., 2007; Sullivan & Prather, 2007) or heterogeneous reactions with polluted air masses (Fitzgerald et al., 2015; Krueger et al., 2004; Sullivan et al., 2009). The chemical change in the physiochemical properties of dust, herein referred to as "aging", can alter the water uptake properties of dust (Krueger et al., 2003; Laskin et al., 2005) thus affecting its light scattering efficiency (Bauer et al., 2007; Kandler et al., 2011a; Levin et al., 1996), cloud droplet activation properties (Gibson et al., 2007; Kelly et al., 2007; Sullivan et al., 2009), and atmospheric lifetime (Abdelkader et al., 2015; Lance et al., 2013; Li et al., 2014; Wu et al., 2013). Chemical aging is also essential to increase the bioavailability of nutrients within dust via ligand-mediated and proton-mediated dissolution processes (Nenes et al., 2011; Spokes & Jickells, 1995; Stockdale et al., 2016). Modeling the chemical aging process is challenging and, as a result, models often treat dust as a chemically homogeneous and hydrophobic particle type (Han et al., 2011; Pringle et al., 2010; Shi et al., 2008), which oversimplifies the complexity of dust and its impacts in the atmosphere.



African dust is the largest source of dust, which is transported to the Caribbean, North
and South America and Europe (Barkley et al., 2019; Prospero & Mayol-Bracero, 2013). Most
research studying African dust transport focuses on the summer when dust mass concentrations
are at a maximum (Prospero & Mayol-Bracero, 2013; Zuidema et al., 2019) and the Saharan Air
Layer (SAL) that transports dust across the Atlantic is at its maximum altitude (Carlson &
Prospero, 1972). The height of African dust transport minimizes the mixing time of transported
dust with the underlying Marine Boundary Layer (MBL) until the dust settles out of the SAL
(Ryder et al., 2018). Studies exploring African dust aging during the summertime have produced
varying results. Early attempts to study dust aging in the western Atlantic from Li-Jones &
Prospero, 1998, found a strong correlation between mineral dust and non-sea salt sulfate (NSS-
$SO_4^{2-}$) from size-resolved filter measurements collected in Barbados during the boreal spring,
which they conclude is the result of dust aging by sulfur emissions from Europe. In more recent
studies, single-particle methods have been utilized to obtain more detail on the mixing state of
dust to unambiguously determine if markers of aging are located within dust or other aerosol
particles. Conclusions from Denjean et al., 2015, inferred from low hygroscopicity growth
measurements, that long range transported (LRT) African dust particles collected over Puerto
Rico in the boreal summer are not chemically processed and are minimally mixed with other
chemical components. In contrast, Fitzgerald et al., 2015 also analyzed LRT African dust in
Puerto Rico during the boreal summer, but found sulfate and oxalate on dust particles using real-
time single-particle mass spectrometry. Dust composition measured by Fitzgerald et al., 2015
may have also been affected by local cloud processing as sampling occurred within a cloud
forest.  Boreal summer analysis of LRT dust in Barbados from Kandler et al., 2018 and
Weinzierl et al., 2017 revealed limited aging of dust particles with some internal mixing with



other chemical components from microscopy and spectroscopy analysis. Model analysis of
summertime African dust transport from Abdelkader et al., 2017 explored the evolution of dust
aging as dust is transported from Africa to the western Atlantic, finding that aged African dust is
quickly removed from the aerosol loading during transit across the Atlantic, leaving minimally-
aged dust within the aerosol loading that reaches the western Atlantic.

Unlike the summertime when dust is transported within the elevated SAL, dust is

transported at lower altitudes during the wintertime (Tsamalis et al., 2013) enhancing the mixing
time of dust within the MBL, which may increase the interaction between dust, anthropogenic
emissions, and marine biogenic emissions in the MBL (Gutleben et al., 2022; Savoie & Prospero,
1982). Additionally, the Sahelian burn season occurs during the boreal winter, often resulting in
long-range co-transport of dust and smoke emissions that were specifically observed during the
ATOMIC/EUREC$^4$A campaigns (Quinn et al., 2021; Royer et al., 2023). Smoke emissions are
known to contain sulfur dioxide ($SO_2$) and nitrogen oxides (e.g., $NOx \equiv NO + NO_2$) that
theoretically could age dust extensively (Hickman et al., 2021; Rickly et al., 2022), especially
during long-range transport in which dust has several days to interact with smoke emissions
before it arrives in the western Atlantic. During the ATOMIC/EUREC$^4$A campaigns, sulfate was
observed on smoke particles, indicating a potential for dust aging during co-transport with smoke
(Royer et al., 2023). However, the likelihood of mixing between the emissions in the MBL, dust,
and African smoke emissions and their impact on dust aging during the wintertime is unclear.

The goal of this study is to determine the extent of aging on African dust particles

transported to the Caribbean during the boreal winter using methods that determine the mixing
state of individual particles. Collection of aerosol samples took place at the Barbados
Atmospheric Chemical Observatory (BACO) at Ragged Point, Barbados from January through



February of 2020 during the ATOMIC/EUREC[4]A campaign. During the sampling period, 3 dust
events consisting of co-transported dust and smoke originating from Africa arrived at Barbados,
offering ample opportunity to study the extent of aging. Lidar measurements also reveal that
dust-laden air masses remained at a low altitude (3.5 km) for the duration of their transit to
Ragged Point, contrasting high altitude summertime transport conditions (Gutleben et al., 2022).
This study provides a unique opportunity for exploring the extent of dust aging on African dust,
whichshould be at a maximum under the conditions described. Our findings  provide much
needed insight into the extent of aging for African aerosols undergoing LRT to the western
Atlantic, which is essential for properly modeling the water uptake properties of dust particles in
the atmosphere as well as the solubility of nutrients in dust bearing minerals.
**3.  Methods**
*3.1 Measurement Site and Sampling Period*

The sampling site, sampling period, and air mass origins that were studied have been

described previously in Royer et al., 2023. Briefly, aerosol samples consisting of long-range co-
transported African dust and smoke as well as marine aerosols were collected at the University
of Miami's Barbados Atmospheric Chemistry Observatory (BACO) at Ragged Point during the
EUREC[4]A and ATOMIC campaigns from 20 January through 20 February 2020 (Quinn et al.,
2021; Stevens et al., 2021). Air mass origins were determined using NOAA's HYSPLIT model
as well as daily dust mass concentrations. Ragged Point (13°6' N, 59°37'W) , a prominence on
Barbados' eastern coast, is an ideal location for studying the extent of dust aging in LRT African
aerosols as it is exposed to the steady easterly trade winds which regularly carry outflows of
African aerosols such as dust and smoke to the island (Archibald et al., 2015; Carlson &



Prospero, 1972; Prospero, 1968) and minimize the influence of anthropogenic activity from local
islands to the west (Prospero et al., 2005; Savoie et al., 2002).

In this study, we utilize both bulk and single-particle methods to highlight the importance

of aerosol mixing state for understanding the extent of dust aging. Bulk methods have
traditionally been used to study long-range dust transport and the extent of dust aging in the
western Atlantic (Chen & Siefert, 2004; Li-Jones & Prospero, 1998; Savoie et al., 2002). Single
particle methods used herein show that elucidating the aerosol mixing state is essential for
determining the full extent of aging in LRT African dust particles.
*3.2 Dust Concentration and Soluble Ion Content*

To collect aerosol samples, the BACO is equipped with a high-volume sampler and an

isokinetic aerosol inlet on top of a 17 m-tall tower situated on a 30 m bluff along the coast at
Ragged Point. Aerosol filters were collected using a high-volume sampler pumping at a rate of
approximately 0.7 $m^3$/min for bulk measurements and 1.0 $m^3$/min for size-resolved
measurements. Size-resolved filters were collected on a cascade impactor (Tisch Environmental,
Inc., Series 230) with 6 stages and 1 backing filter. Size-resolved aerosol analysis are separated
into supermicron (stages 1-4; >1.3 μm) and submicron (stages 5 – backing filter; <1.3 μm). All
aerosol filter samples were collected on cellulose Whatman-41 (W-41) filters with a nominal 20
μm pore size. After filter collection, filters were washed with Milli-Q water three times to
remove soluble material. The washed filters were subsequently combusted in a furnace at 500 °C
for about 12 h (i.e., overnight) to remove the cellulose filter and determine daily dust mass
concentrations (Prospero et al., 2021; Zuidema et al., 2019). Procedural filter blanks were also
collected by placing a filter in the sampler cassette for 15 min without turning on the pump. The
resulting ash mass from a sample minus the mass of a filter blank is the gross ash weight, which



is then adjusted by a factor of 1.3 to convert the ash weight to a mineral dust concentration
(Prospero, 1999; Zuidema et al., 2019).

To determine daily bulk and size-resolved soluble ion content, the 20 mL of Milli-Q used

to wash the filters was filtered through a 25 mm membrane filter with 0.4 um pore size
(Whatman Nuclepore Track Etch Membrane) to remove any particulates from the washing
process. Filtrate was then frozen in a -20°C freezer until analysis. To prepare samples, frozen
filtrate was thawed in a warm water bath and vortexed for 20 sec. The filtrate was then analyzed
using an ion chromatography (IC) instrument (Dionex Integrion HPIC System; Thermo
Scientific). Samples were analyzed in triplicate to ensure precision of results. To obtain soluble
ion content, 5 mL aliquots of filtrate were injected into the IC system and analyzed for cations
(IonPac CG12A/CS12A; Thermo Scientific) and anions (IonPac AS11-HC; Thermo Scientific).
Cations of interest analyzed by IC include lithium ($Li^+$), sodium ($Na^+$), ammonium ($NH_4^+$),
potassium ($K^+$), magnesium ($Mg^+$), and calcium ($Ca^+$) while anions of interest include fluoride
($F^-$), formate ($CH_2O^-$) , methanesulfonate (MSA), chloride ($Cl^-$), nitrite ($NO_2^-$), bromide ($Br^-$),
nitrate ($NO_3^-$), sulfate ($SO_4^{2-}$), oxalate, and phosphate ($PO_4^{3-}$). Since the filtrate analyzed includes
sea salt emissions, which may include sulfate from ocean emissions, non-sea salt sulfate (NSS-
$SO_4^{2-}$) was calculated using the equation
$$[NSS - SO_4^{2-}] = [SO_4^{2-}] - (0.2517 * [Na]) \tag{1}$$
to determine the fraction of sulfate derived from non-sea spray emissions including
anthropogenic and marine biogenic sources. For the purposes of this study, we focus primarily
on NSS-$SO_4^{2-}$ and $NO_3^-$ as they are chemical indicators of dust aging (Sullivan et al., 2007) as
well as oxalate as it indicates cloud processing and other processes (Ma et al., 2004). Samples
were also analyzed for methanesulfonate, an important tracer of ocean-derived biogenic sulfur





(Gaston et al., 2010) that can age dust particles (Desboeufs et al., 2024), however MSA
measurements were negligible.
*3.3 Aerosol Mixing State Analysis*

To determine the mixing state of individual particles, aerosol samples were collected at

ambient relative humidity (RH) through an isokinetic aerosol inlet with a three-stage
microanalysis particle sampler (MPS-3, California Measurements, Inc.), which samples particles
from diameters of 5.0-2.5 μm (stage 1), 2.5-0.7 μm (stage 2), and <0.7 μm (stage 3). For each set
of samples (one set including one sample from each stage of the MPS), the MPS was run for 45
min at 2 L/min flow starting at approximately 09:30 LT (local time) or 13:30 coordinated
universal time (UTC). Meteorological data from a local station were used to manually check that
wind direction fell between 335 and 130 degrees and that wind speeds were greater than 1 m/s
during all sampling periods to ensure that only air from the open ocean to the east was sampled
rather than local, anthropogenically-influenced air.
3.4 *Single Particle Elemental Composition*

To determine aerosol elemental composition, particles were deposited onto carbon-coated

copper grids (Ted Pella, Inc., Prod. # 01910-F) on each of the 3 stages of the MPS that were later
analyzed at the Environmental Molecular Science Laboratory (EMSL) at Pacific Northwest
National Laboratory (PNNL) using computer-controlled scanning electron microscopy (Quanta
3D) coupled with energy-disperive X-ray spectroscopy (EDAX, Inc.) (CCSEM/EDX).
Approximately 1800 particles from stage 1, 2500 particles from stage 2, and 3200 particles from
stage 3 were analyzed via CCSEM/EDX for each day of sampling. Only particles with diameters
>0.1 μm were analyzed. Semiquantitative data products from CCSEM/EDX analysis were then
analyzed in MATLAB (version 9.6.0; The Mathworks, Inc.) using a k-means clustering



algorithm (Ault et al., 2012; Royer et al., 2023; Shen et al., 2016) to group similar particles into
clusters based on the elemental percentage, size, and shape of individual particles. These clusters
are then assigned to particle types based on their morphology, characteristic EDX spectra, and
the existing literature. Percent composition threshold values of 1% were used when processing
CCSEM/EDX data to ensure the presence of elements detected by the EDX. Single particle
analysis using CCSEM/EDX was limited to 16 elements found in common aerosols such as dust,
sea salt, and smoke particles: carbon (C), nitrogen (N), oxygen (O), sodium (Na), magnesium
(Mg), aluminum (Al), silicon (Si), phosphorus (P), sulfur (S), chlorine (Cl), potassium (K),
calcium (Ca), vanadium (V), manganese (Mn), iron (Fe), and nickel (Ni). To prevent the signal
from the Cu-grid from interfering with the particles, Cu was intentionally excluded from the list
of elements to detect. Though the grids are coated with C, C was intentionally included due to
the presence of organics within the aerosol loading. The inclusion of C may result in a
suppression of other elements of interest as the C signal may be artificially elevated by the C-
coating on the Cu-grids.

Particle identification was primarily based on semiquantitative elemental composition

determined by EDX. Dust particles were identified based on the presence of elements common
in aluminosilicate minerals, including Si, Al, Fe, K, Ca, and Mg detected from EDX analysis
(Hand et al., 2010; Krueger et al., 2004; Levin et al., 2005). Sea salt particles were characterized
by high Na and Cl content, indicating the presence of halite (NaCl). Internal mixtures of dust and
sea salt contained elements indicative of both dust (Si, Al, Fe, K, Ca, and Mg) and sea salt (Na
and Cl), usually with portions of the particle containing primarily dust with other portions
containing primarily sea salt. Internally mixed dust and smoke contained elements common in
mineral dust as well as high C, K, and S which are representative of carbon-based matter that has





undergone combustion and aging from sulfur componds, leading to the formation of potassium-
containing salts (Andreae, 1983; Li et al., 2003).

Analysis of CCSEM/EDX data also included calculating the extent of aging across the

aerosol size distribution. To obtain this information, N and S % values for each particle in a
cluster of known particle type were extracted along with the corresponding diameter for each
particle. Data was then binned according to diameter size, while N and S % values were
averaged for each diameter size. Values that did not exceed 1% were rounded down to 0% as
only an exceedance of 1% guarantees the presence of an element.

The spatial distribution of elemental components on select particles was also determined

using elemental mapping (AZtecLive SmartMapping; Oxford Instruments). Approximately 10
elemental maps were collected for dust, sea salt, and internally mixed dust and sea salt. Spectra
were collected for select components within elemental maps to obtain more detailed chemistry
across an individual particle, which allows for analysis of sea salt and dust components as well
as the extent of aging in these components within internally mixed dust and sea salt particles.
3.5 *Aerosol Surficial Chemical Composition*

The spatial distribution of major ions across the surface of individual particles was also

determined. Aerosol particles were collected onto silicon wafers (Ted Pella, Inc., Prod. # 16008)
within the MPS, which were then analyzed with time-of-flight secondary ion mass spectrometry
(TOF-SIMS; IONTOF GmbH, Munster, Germany) at PNNL (Li et al., 2023). In addition to
providing more detailed chemical information on dust aging, TOF-SIMS supplements the time-
intensive method of elemental mapping with SEM/EDX by analyzing multiple particles to
explore particle aging, thus supporting the representativeness of elemental mapping results to the
total aerosol loading (Hopkins et al., 2008). Further, while EDX is limited to elemental data



indicative of aging (e.g., the presence of N and/or S), TOF-SIMS can detect compounds such as
nitrate ($NO_3^-$) and sulfate ($HSO_4^-$) ions that more concretely provide evidence of chemical aging.

To perform TOF-SIMS analysis, a 25 keV $Bi_3^+$ beam was focused to around a 0.4 um

diameter area on the silicon substrate and scanned over a 100 μm x100 $μm^2$ area to produce an
image of 256x256 pixels. The current of the beam was 0.36 pA with 10 kHz pulse frequency,
and data collection time was 600 s per set of images. The total ion dose for each sample was
under the static limit so that only surface information (<2 nm) was collected for the analyzed
particles. Delayed extraction mode was also used during image collection to ensure that both
positive and negative ion images could be collected at the exact same location. Ions of interest
indicated the presence of sea salt ($Na_2Cl^+$ and $NaCl_2^-$), dust ($Al^+$ and $Ca^+$), and chemical aging
($HSO_4^-$ and $NO_3^-$). Surface contamination from the lab space the samples were handled in (e.g.,
butanediol (m/z -89) likely from butanol used in particle counters in the lab space) was observed
in the samples, and to remove contamination, a 20 keV argon (Ar) cluster ($Ar_{1500\pm300}^+$) sputtering
ion beam was used with a beam current of about 2.0 nA before chemical analysis with the 25
keV $Bi_3^+$ beam occurred. Samples underwent Ar sputtering for 50s to remove the top 100 nm of
sample.
4. **Results**
*4.1 Bulk Aerosol Analysis*

Figure 1 presents daily dust mass concentrations and bulk soluble ion content along with

correlation plots for each ion of interest. Results show a strong correlation between daily dust
mass concentrations and nitrate ($NO_3^-$, $R^2 = 0.75$) and non-sea salt sulfate (NSS-$SO_4^{2-}$, $R^2 =$
0.83), as well as a weak correlation with oxalate ($R^2 = 0.11$) throughout the entire campaign. The
presence of nitrate, non-sea salt sulfate, and oxalate from bulk aerosol analysis suggests that dust





is being aged during transport. Similar figures for supermicron and submicron analysis can be
found in Figures S1 and S2, respectively, in the Supporting Information (SI). Notably, dust mass
concentrations were evenly split between the supermicron and submicron size modes. Both
supermicron (*SUP*) and submicron (*SUB*) analysis shows similar findings to bulk filter analysis
in which a strong correlation exists between dust mass concentrations and nitrate ($R^2_{SUP\ NO3}$ =
0.50; $R^2_{SUB\ NO3}$ = 0.59), non-sea salt sulfate ( $R^2_{SUP\ NSS-SO4}$ = 0.80; $R^2_{SUB\ NSS-SO4}$ = 0.49), and
oxalate ($R^2_{SUP\ oxalate}$ = 0.63; $R^2_{SUB\ oxalate}$ = 0.27). Differences between supermicron and submicron
analysis indicate that nitrate is more concentrated in the supermicron aerosol loading ($Avg_{SUP\ NO3}$
= 0.64 µg/m$^3$; $Avg_{SUB\ NO3}$ = 0.44 µg/m$^3$). However, nitrate has an appreciable submicron mode
likely due to transported African smoke. Results also show that $NSS-SO_4^{2-}$ ($Avg_{SUP\ NSS-SO4}$ =
0.04µg/m$^3$; $Avg_{SUB\ NSS-SO4}$ = 0.46µg/m$^3$) and oxalate ($Avg_{SUP\ oxalate}$= 0.01µg/m$^3$; $Avg_{SUB\ oxalate}$ =
0.04µg/m$^3$) are more concentrated in the submicron aerosol loading (Quinn et al., 2021; Royer et
al., 2023; Savoie et al., 1982). These findings suggest that dust is possibly being aged, with
supermicron dust being primarily aged by nitrate and submicron dust being primarily aged by
sulfate and oxalate or via cloud processing (Bondy et al., 2017). However, the limitations of this
traditional analysis include an oversimplification of aerosol mixing state by assuming nitrate,
non-sea salt sulfate, and oxalate are associated only with dust.
*4.2 Size-Resolved Aerosol Mixing State*

Using size-resolved chemical data of individual particles from CCSEM/EDX analysis,

we assessed the extent of aging across the aerosol size distribution. The role of particle size is
important as smaller dust particles have higher surface area-to-volume ratios that have been
suggested to increase the propensity for dust aging (Baker & Jickells, 2006). Results from
CCSEM/EDX analysis revealed the presence of both marine particles such as sea spray, aged sea



spray, organics, and sulfates, as well as continental particle types including dust, internally
mixed dust and sea salt, internally mixed dust and smoke, and smoke that are described in detail
in Royer et al., 2023. Here, we focus on 4 particle types relevant to the understanding of dust
aging during the sampling period: dust, sea spray (combined sea spray and aged sea spray),
internally mixed dust and sea salt, and internally mixed dust and smoke.
Figure 2 presents detailed size-resolved chemical data for the four particle types of
interest during periods of dust transport to Barbados, providing insight into the extent of particle
aging across the aerosol size distribution. In Figure 2a, particle types are plotted as number
fractions of the aerosol loading as a function of aerosol diameter. Number fractions of dust
particles determined from CCSEM analysis contrast dust mass concentrations determined from
bulk aerosol analysis. Dust mass concentrations for the submicron and supermicron aerosol
loading are similar at $14.38\mu g/m^3$ and $12.28\mu g/m^3$, respectively. However, CCSEM analysis
reveals a large difference in the number fractions of submicron and supermicron dust particles,
where dust only makes up 21% and 4% of the submicron and supermicron aerosol loading by
particle number, respectively. This discrepancy is not only due to the difference in individual
particle mass between sub- and supermicron particles, but also due to the oversimplification of
dust particles in bulk analysis. Figure 2a reveals that internal mixtures of dust with other
components such as sea salt and smoke are, in fact, more abundant in the aerosol size distribution
compared to dust alone, specifically in the supermicron aerosol loading. Internal mixtures of dust
and sea salt comprise 11% of the submicron and 45% of the supermicron aerosol loading, while
internal mixtures of dust and smoke comprise 14% of the submicron and 8% of the supermicron
aerosol loading. These data reveal a complexity in the aerosol loading overlooked by bulk


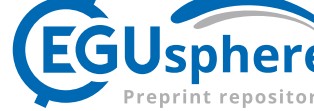

aerosol analysis, and call into question the assumption that the presence of nitrate, non-sea salt
sulfate, and oxalate in bulk samples indicates aging of individual dust particles.

Figures 2b, c, d, and e present the average detectable N content (left axis) and number

fraction of N-containing particles (right axis) as a function of size for sea salt, internally mixed
dust and sea salt, internally mixed dust and smoke, and dust particles, respectively. Values for
average N content and particle fractions with detectable N for each particle type are reported in
the Supporting Information (SI) tables S1 and S2, respectively. Notably, average N content
within individual particles is similar for all 4 particle types, ranging from 2.0±0.5% to 2.6±0.4%
N with an average N content of 2.2±0.6%. However, between the 4 particle types, there are large
variations in the number of particles containing detectable N. Detectable N is found in 36±25%
of submicron and 76±12% of supermicron dust particles. This corroborates findings from bulk
aerosol analysis which shows that nitrate is more concentrated in the supermicron aerosol
loading compared to the submicron aerosol loading. However, figure 2a reveals that dust only
comprises 13% of the aerosol loading by number, with a much larger contribution in the
submicron aerosol loading. Considering that individual particles containing detectable N on
average have a N content of 2.2±0.6%, dust alone cannot explain trends in bulk nitrate. Figure
2c, d, and e all indicate the presence of N in particles aside from dust. Most notably, internally
mixed dust and sea salt, which makes up 45% of the supermicron aerosol loading, has detectable
N in 75±14% of supermicron particles, potentially explaining the high supermicron nitrate
content from bulk aerosol analysis.

Figures 2f, g, h, and i present the average detectable S content and number fraction of S-

containing particles for the 4 particle types of interest. Values for average S content and particle
fractions with detectable S for each particle type are reported in SI tables S3 and S4,



respectively. Similarly to Figures 2b, c, d, and e, the S plots indicate that S content in individual
particles does not vary much across the aerosol size distribution for each particle type, with
average S content for the total aerosol loading ranging from $1.7\pm0.5\%$ to $2.7\pm0.5\%$ and
averaging $2.4\pm1.2\%$. However, particle fractions with detectable S do vary across the aerosol
size distribution and between particle types. Only $8\pm7\%$ of submicron and $14\pm15\%$ of
supermicron dust particles contained detectable S, indicating a large discrepancy between dust
particles containing S and the S mass concentrations observed. For almost all particle types
across the aerosol size distribution, with the exception of supermicron internal mixtures of dust
and smoke, S content exceeds that of dust. Internal mixtures of dust and sea salt far exceed the S
content in dust with $28\pm9\%$ of submicron particles and $35\pm19\%$ of supermicron particles
containing S. This is especially relevant as internal mixtures of dust and sea salt comprise a
much larger portion of the supermicron aerosol loading at 45%, compared to dust at 4%. Sea salt
particles, which also make up a much larger portion of the particle loading than dust at 55% of
the submicron and 44% of the supermicron aerosol loading, also contained larger number
fractions of particles with detectable S where $36\pm22\%$ of submicron and $87\pm58\%$ of supermicron
sea salt particles had observable S. However, an increase in S in sea salt particles across the size
distribution is likely the result of calcium-sulfate minerals that often form on larger sea salt
particles (Ault et al., 2013; Bondy et al., 2018; Choël et al., 2007). Observed S in sea salt and
internal mixtures of dust and sea salt may explain the observed non-sea salt sulfate mass
concentrations in the supermicron size range. However, number fractions of particles containing
detectable S were greater in the supermicron size range compared to the submicron size range for
all four particle types except internally mixed dust and smoke. Internally mixed dust and smoke
only comprises 14% of the submicron aerosol loading and has an average S content of $2.7\pm1.0\%$



for submicron particles. Instead, high mass concentrations of sulfate observed in submicron IC
analysis are likely caused by ammonium sulfate particles observed during the sampling period.
Previous studies show sulfate particles are primarily submicron in size and comprise a large
fraction of the submicron aerosol loading during both clean marine conditions and African dust
transport, which may contribute high non-sea salt sulfate mass detectable in bulk aerosol analysis
but not in S content for the four particle types of interest (Royer et al., 2023).

Ternary plots presented in Figure 3 provide insight into the chemical mixing state of

detected particles during the EUREC[4]A/ATOMIC campaign. Single particles are represented by
individual dots in each ternary plot. The color of each dot indicates the diameter of the particle it
represents. The position of each dot within the ternary plot indicates the relative abundance of S
(left vertex), Cl (top vertex), and N (right vertex) within each particle. Sea salt particles
displayed in Figure 3a are clustered primarily at the top vertex indicating high Cl content, low S
content, and low N content characteristic of freshly emitted sea salt. Particle clustering at the left
and right vertices demonstrates aging of sea salt particles as low Cl content and high S and/or N
content suggests chloride has been depleted and replaced with nitrate and sulfate during
heterogeneous reactions (Ault et al., 2013, 2014; Behnke et al., 1997; Gaston et al., 2011, 2013;
Sobanska et al., 2003). Differences in color at each vertex within Figure 3a also indicates that
aged sea salt particles are smaller than fresh sea salt particles on average (Laskin et al., 2012).

Dust and internally mixed dust and smoke particles are primarily in the submicron size

range, as indicated in Figure 2, and cluster at the right vertex indicating the presence of more
nitrate compared to sulfate and chloride in these particles, but these particles are sparse in
number concentration. Internally mixed dust and sea salt particles cluster primarily along the
right axis of the ternary plot, exhibiting various levels of Cl likely from sea salt, low S, and



varying N content. Notably, larger particles identified as internally mixed dust and sea salt had
elevated chloride compared to S and N while smaller particles had less chloride and elevated S
and N. Figure 3 demonstrates that, of the particle types containing dust components, internal
mixtures of dust and sea salt are the most abundant in the aerosol loading and had the highest
number of aged particles containing detectable N and S. As such, we focus our analysis of
chemical aging on internally mixed dust and sea salt.
*4.3 Elemental Mapping*

While analysis by CCSEM/EDX is valuable for determining size-resolved aerosol mixing

state, this method does not map the distribution of elements within each particle, which leads to
uncertainty regarding the location of the aging components within the particles, particularly for
internally mixed particle types. The elemental mapping image depicted in Figure 4a shows the
distribution of elements across an entire particle. The presence of distinct areas of dust
components such as Si, Al, Ca, Fe, and Mg that are separate from sea salt components such as
Na and Cl within a single particle indicates that the particle is a typical internally mixed dust and
sea salt particle. Within the image, the distribution of nitrogen and sulfur are depicted as well,
and visually appear present only over the sea salt components. Spectra from EDX analysis were
extracted from these distinct regions on the particle which reveal that nitrogen and sulfur are
indeed either negligible or completely absent on the dust components (S = 0.1%; N = 0%), while
they are present in appreciable quantities on the sea salt components (S = 2.9%; N = 2.9%). Also
worth noting is the absence of Cl in the sea salt component, which is indicative of sea salt aging
also observed in CCSEM/EDX analysis. This particle is representative of internally mixed dust
and sea salt particles detected during the sampling period that similarly show only aging on the
sea salt components. Additional examples of internally mixed dust and sea salt particle elemental





maps and EDX spectra are provided in Figure S3 of the SI. In addition, elemental maps and
corresponding EDX spectra for dust, sea salt, smoke, and internally mixed dust and smoke are
provided in Figure S4. Similar to the internally mixed dust and sea spray particles, externally
mixed dust and the dust component that is internally mixed with smoke show a lack of aging
while the smoke components in mixed dust and smoke particles show extensive accumulation of
sulfate. These results suggest that even in internal mixtures of dust, only the sea salt (or smoke)
components are undergoing ageing while dust is unprocesed
*4.4 TOF-SIMS Imaging*

Results from TOF-SIMS analysis corroborate findings from SEM/EDX elemental

mapping, indicating that aging of internally mixed dust and sea salt particles occurs primarily on
the sea salt components. Figure 5 depicts results from TOF-SIMS imaging in which the color
intensity in each image represents the intensity of an ion. The cation images in the top panel
indicate the presence of sea salt ($Na_2Cl^+$) and dust ($Al^+$ and $Ca^+$), while the last cation image on
the righthand side shows all ions plotted together. The cation image indicates that while the
majority of the particles in the image presented are sea salt particles due to the abundance of
$Na_2Cl^+$, the co-location of sea salt components with dust components $Al^+$ and $Ca^+$ suggest there
is internal mixing of dust and sea salt as well. The anion images in the bottom panel similarly
plot individual ions, with the anions indicating the presence of sea salt ($NaCl_2^-$) and chemical
markers of aging from sulfate and nitrate ($HSO_4^-$ and $NO_3^-$, respectively) as well as a final image
containing all anions plotted together. Once again, the presence of the anion $NaCl_2^-$ suggests that
sea salt is abundant and further corroborates the cation images. The presence of $HSO_4^-$ and $NO_3^-$
provide insight into the extent of aging on the particles presented in the images which show a
strong presence of aging from sulfate through the presence of $HSO_4^-$ but a lack of aging from



437 nitrate through the absence of $NO_3^-$. Most notably, in the image overlaying all anions together,

438 the $NaCl_2^-$ and $HSO_4^-$ are indistinguishable from one another, indicating aging on the sea salt

439 components. Comparing the cation image to the anion image, it is clear that the areas in which

440 dust components are present are not undergoing aging, rather, primarily the sea salt components

441 are being aged. This supports findings from elemental mapping from SEM/EDX analysis which

442 similarly suggest minimal aging of dust components but aging of sea spray components in these

443 internally mixed particles.

444 **Discussion & Conclusion**

445  Traditional methods for studying dust aging often measure dust concentrations (or their

446 proxies) and soluble materials extracted from aerosol filters that oversimplify the aerosol

447 loading. Previous studies have historically used correlations between nss-sulfate, oxalate, and

448 nitrate and dust mass concentrations to prove the presence of dust aging, but have been unable to

449 determine the mixing state of dust and, thus, whether the dust is actually undergoing aging with

450 traditional methods (Chen & Siefert, 2004; Li-Jones & Prospero, 1998). The results from this

451 work indicate that while internal mixtures of dust with other particles are common in the lower

452 boundary layer, both internally and externally mixed African dust detected in the western

453 Atlantic are minimally aged during the wintertime. The boreal winter provides the most ideal

454 conditions for African dust aging to occur due to the lower transport altitude creating more time

455 for MBL emissions to interact with LRT dust. Further, wintertime dust is often co-transported

456 with Sahelian biomass burning emissions contributing high concentrations of NOx and $SO_2$ that

457 are co-transported with dust, allowing for interaction of dust with aging components over several

458 days during its transit to the western Atlantic (Hickman et al., 2021). The lack of aging on

459 internal mixtures of dust and smoke indicate rapid conversion of $NO_x$ and $SO_2$ to nitrate and



sulfate on smoke, which is corroborated by the presence of potassium sulfate salts observed on
smoke particles from EUREC$^4$A/ATOMIC and a lack of these compounds on dust (Royer et al.,
2023). The lack of dust aging indicators observed during the winter may indicate a lack of aging
throughout the year for LRT African dust. This is apparent from size-resolved CCSEM/EDX
data which indicate a lack of aging on dust particles by sulfate and nitrate, and from elemental
mapping and TOF-SIMS imaging which show that chemical aging is favored on the sea spray
and smoke components of internally mixed dust particles.

It is likely that dust is being aged in the eastern Atlantic, but is removed during LRT to

the western Atlantic based on previous studies (Abdelkader et al., 2017). Further, it is likely that
any aging that occurs on dust in the eastern Atlantic is removed during LRT to the western
Atlantic. The lower altitude for dust transport during the wintertime would also provide ample
opportunity for dust aging in the eastern Atlantic which could lead to the rapid aging of dust
particles before LRT (Chiapello et al., 1995; Kandler et al., 2011; Ullerstam et al., 2002). The
addition of high levels of pollutants from the Sahelian fires could also exacerbate aging in the
eastern Atlantic (Andreae et al., 2000). Aged dust particles are much more efficiently removed
by both wet and dry deposition as well as cloud droplet activation as a result of increased water
uptake properties enhancing their size and reactivity (Abdelkader et al., 2017; Gaston, 2020;
Metzger et al., 2006). It is possible that the rapid aging of dust particles in the eastern Atlantic
upon dust emission increases the water uptake properties of the dust, leading to rapid removal of
dust particles before LRT can carry dust particles to the western Atlantic. This potentially
explains the lack of aging on dust particles in the western Atlantic, as any aged dust is likely
removed before arriving over Barbados.



Based on the high abundance of internally mixed dust and sea salt particles in the aerosol
loading, it is likely that unaged dust transported across the Atlantic becomes associated with
aged sea spray as dust is being detrained from from the SAL into the MBL as observed in
previous studies (Abdelkader et al., 2017). This would result in altitudinal gradients in dust
mixing state important for dust radiative impacts. Though dust components in the lower
boundary layer are rarely aged, internal mixtures of dust with other components such as sea salt
and smoke are common. The high degree of aging on these internally mixed components suggest
internal dust mixtures are more hygroscopic and thus are potentially efficient as cloud
condensation nuclei.
The lack of aging on dust components has implications for nutrient availability in mineral
dust aerosols transported to the tropical Atlantic as well. Ecosystems in the tropical Atlantic such
as the open Atlantic Ocean and the Amazon rainforest rely on external inputs of nutrients such as
iron and phosphorus (Fe and P). Chemical aging is particularly important to provide bioavailable
sources of nutrients to marine ecosystems, as deposition of particles out of the euphotic zone
competes with nutrient release into seawater (Gaston, 2020). The lack of aging on mineral dust
observed in this study suggests African dust may not be as important of a source of bioavailable
nutrients for the North Atlantic as previously assumed.
In this work, we utilized methods that target aerosol mixing state to determine the extent
of aging in LRT African dust particles to the western Atlantic during the wintertime. The
disparity between bulk methods, which suggest dust aging is extensive, and methods that
characterize the aerosol mixing state, which reveal a distinct lack of aging for dust components,
reveals the importance of utilizing single-particle methods to understand dust aging (Fitzgerald et
al., 2015; Kandler et al., 2011). We also provide much-needed insight into the question of dust



aging in the western Atlantic, revealing a lack of aging for dust particles in the wintertime that
should be considered in global and regional models.
**Data Availability**
The data will be publicly available in the University of Miami data repository.
**Author Contribution:**

Conceptualization of this was performed by HMR, APA, and CJG. Collection of samples was
conducted by HMR, while analysis was done by HMR, MS, HE, NNL, ZC, and ZZ. Development
of methods used in this work was done by HMR, ZC, SC, APA, and CJG. Instrumentation used to
conduct this work was provided by CJG, APA, SC, and ZZ. Validation of data products was
performed by HMR, ZC, APA, and CJG. Computer code used for data analysis was provided by
APA. Data visualization was performed by HMR, APA, and CJG. Supervision and project
administration duties were conducted by CJG. CJG is solely responsible for funding acquisition.
HMR wrote the original draft for publication, and all co-authors reviewed and edited this work.
**Competing Interests:** The authors declare that they have no conflict of interest.

**Acknowledgements**
C.J.G. acknowledges an NSF CAREER award (1944958). A portion of this research was
performed on project awards (10.46936/lser.proj.2019.50816/60000110 and
10.46936/lser.proj.2021.51900/60000361) from the Environmental Molecular Sciences
Laboratory, a DOE Office of Science User Facility sponsored by the Biological and
Environmental Research program under Contract No. DE-AC05-76RL01830.


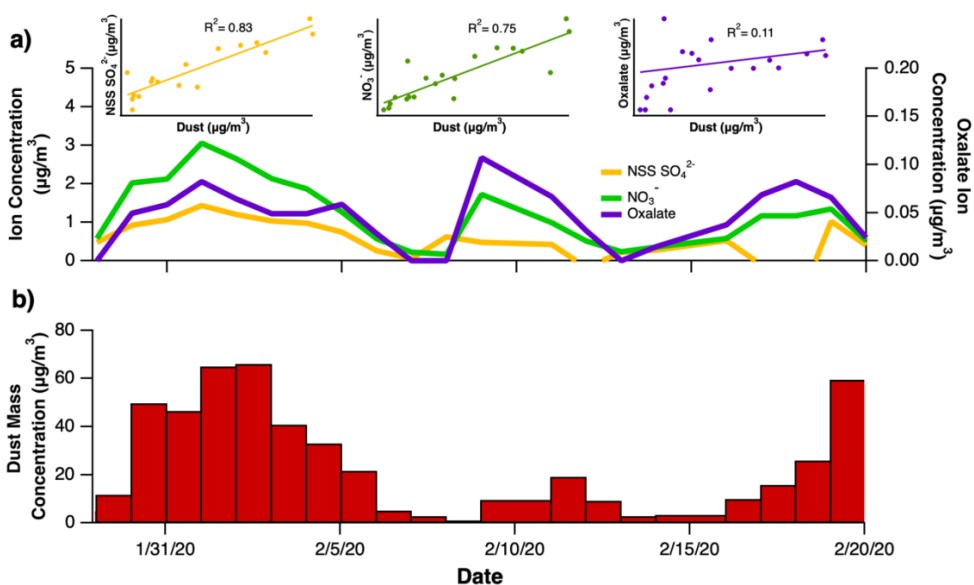


Figure 1 – Daily bulk soluble ion content for nitrate (left axis; green line), non-sea salt sulfate
(left axis; yellow line), and oxalate (right axis; purple line) with correlation plots for each ion as
a function of dust mass concentrations a) and daily bulk dust mass concentrations b) determined
for the entire sampling period. Correlation plots include all data, including data from samples
with undetectable ions, but trendlines only consider data with detectable dust and ions.





Figure 2 – Size-resolved chemistry plots summarized by a) a total size-resolved chemistry plot in

which the particle number loading is normalized to the sum of sea salt particles (blue), internally

mixed dust and sea salt particles (purple), internally mixed dust and smoke particles (brown), and



dust particles (red) for each size bin and presented as a fraction of the particle number loading in
each size bin (left axis) along with the total sum of the number of particles of interest for each
size bin (right axis: black line). Plots depicting the average N or S content in individual particles
(left axis; thick line) as well as the number fraction of particles in each size bin containing N or S
(right axis; thin line) are provided for b) N in sea salt particles, c) N in internally mixed dust and
sea salt particles, d) N in internally mixed dust and smoke particles, e) N in dust particles, f) S in
sea salt particles, g) S in internally mixed dust and sea salt particles, h) S in internally mixed dust
and smoke particles, and i) S in dust particles.

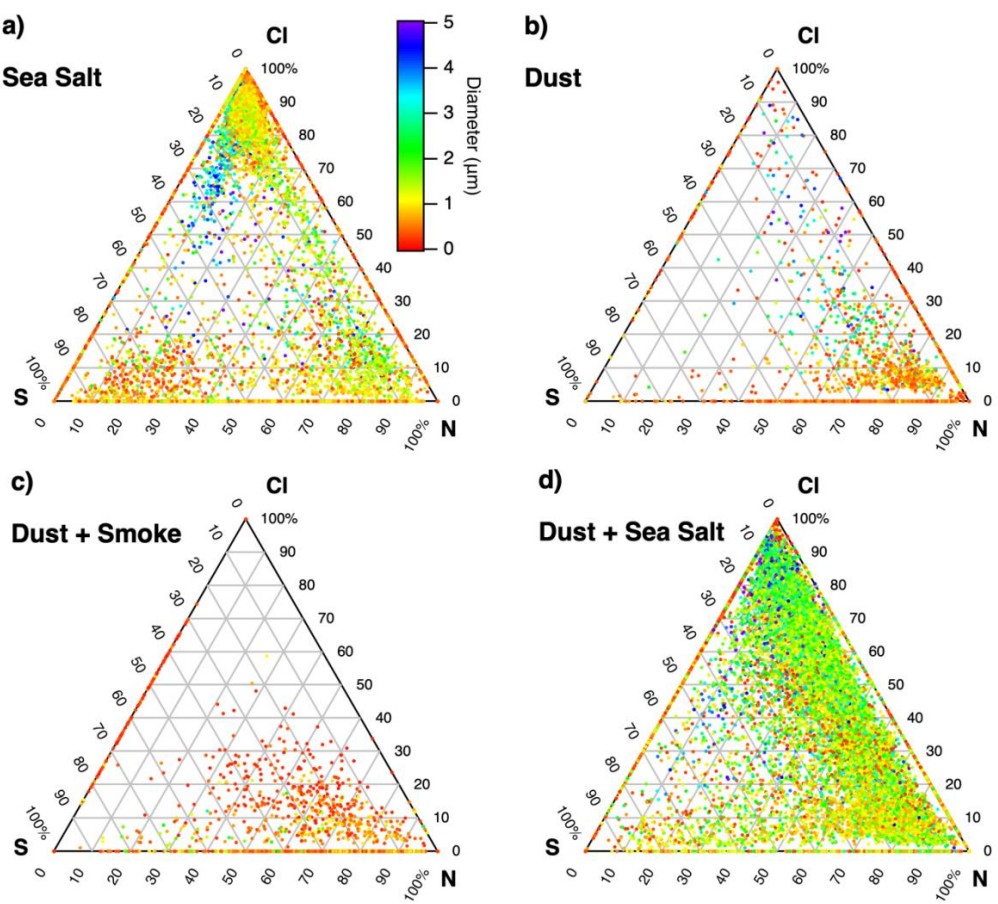


Figure 3 – Ternary plots presenting the normalized percentage of S (left axis), Cl, (right axis),

and N (bottom axis) present in dust (6426 particles), internally mixed dust + smoke (N=1588 pt),

internally mixed dust + sea salt (18,210 particles), and sea salt (22,354 particles). Color scaling

denotes particle diameter.

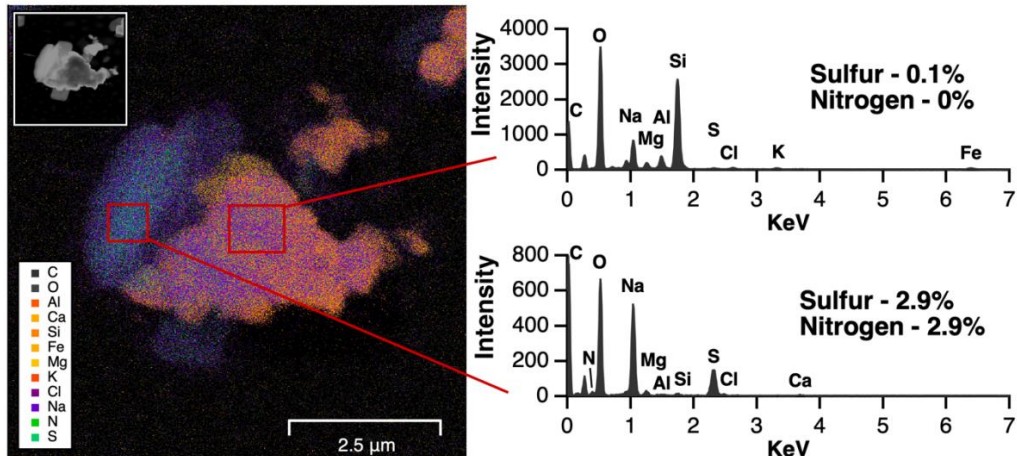


Figure 4 – Elemental mapping image from SEM/EDX analysis for an internally mixed dust and
sea salt particle collected at Ragged Point on 2/9/2020 on stage 2 of the MPS. Top left plot
depicts the SEM image. The legend explains the color associated with each element plotted in
the elemental map with warm colors denoting dust components, cool colors denoting sea spray
components and green colors denoting aging markers (a). Red squares on the elemental map
indicate where EDX spectra were extracted for the dust component (b) and the sea salt
component (c) of the particle. Sulfur and nitrogen values represent calculated EDX intensity
present on dust and sea spray components of the particle.



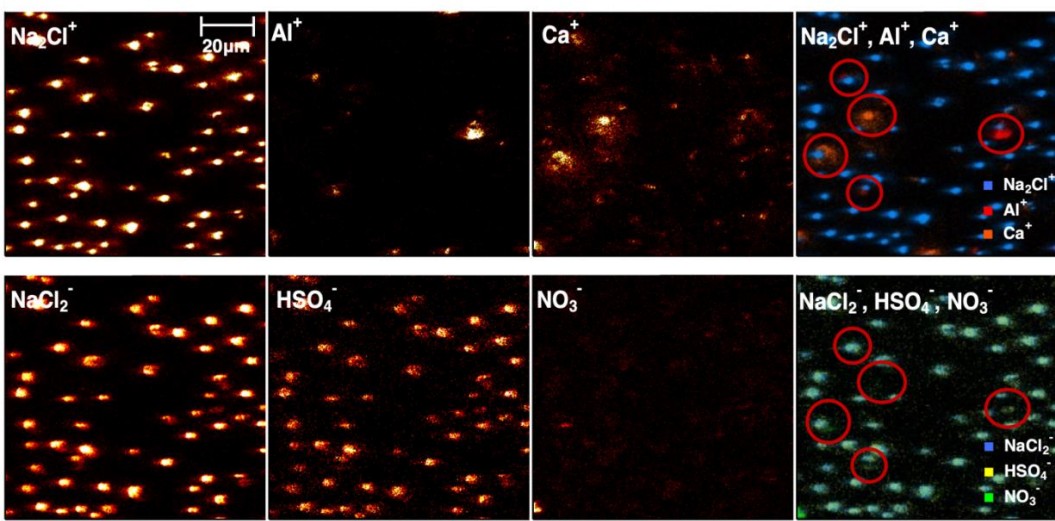


Figure 5 – Image plots from TOF-SIMS analysis of a sample collected on 2/18/2020 from stage

1 of the MPS. Red circles mark the location of dust components of the particles.




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
