# Peer review of "African dust transported to Barbados in the Wintertime Lacks Indicators of Chemical Aging Haley M. Royer1,2, Michael T. Sheridan1,3, Hope E. Elliott4, Edmund Blades1, Nurun Nahar Lata5, Zezhen Cheng5, Swarup China5, Zihua Z"

_EGUsphere, 2024_

## Author Response (AR1)

**Author Responses to Reviewer Comments are in Red

REVIEWER #1

Royer et al. used bulk and single particle analysis to investigate African dust particles transported to Barbados. Although bulk analysis seems to imply significant aging of dust particles after long range transport, single particle analysis suggested that most dust particles were not aged after arrival at Barbados. This work provides novel insights into the mixing state of dust particles after long range transport, and has important implications for their climatic and biogeochemical impacts. The manuscript, which I found very interesting, can be accepted after some minor revision.

We thank the reviewer for their helpful comments.

**Comments:**

Line 38: It can be great to add 1-2 sentences at the end of the abstract to discuss the broad implications of the results this work reported.

We thank the reviewer for this excellent suggestion. We have added the following text:

Lines 40-45: "Our results suggest that chemical aging may only modestly increase the solubility of nutrients in African dust during long range transport. Because most dust that we measured was internally mixed with sea salt, chemical aging is not necessarily required to increase the hygroscopicity of dust, at least in the lower boundary layer. Further, our findings have implications for understanding the release of halogens from sea salts, which may be enhanced in internally mixed dust and sea salt particles."

We have also added a sentence about halogen liberation from sea salt in the Discussion.

Lines 490-493: "Our findings of aged sea spray, with losses of chloride, on internally mixed dust and sea spray particles could also indicate an increase in the heterogeneous displacement of chloride from these internally mixed particles compared to sea spray alone (van Herpen et al., 2023)."

Line 63: Tang et al. (2016) reviewed the effects of chemical aging on CCN activities of mineral dust, and the authors may also cite this paper.

We have added the suggested reference.

Line 86: I would suggest using long range transported instead of "LRT" in the manuscript, as "LRT" really reduces the readability.

We have changed the text as suggested.

Line 179: Please add a reference for this equation.

We now reference Gaston et al., 2024 for the equation to calculate non-sea salt sulfate mass concentrations from our filters.

Line 311: Our work (Zhang et al., 2023) found that similar Al contents in fine and coarse particles for Asian dust, supporting what this work found for African dust.

We have noted the similarity of our results for African dust to Asian dust and have added the suggested reference.

Line 324-373: Could the authors estimate the thresholds (mass fraction in %) when nitrate and sulfate became undetectable? Such information will provide a rough idea about the upper limit of nitrate and sulfate contents when they were not detected.

The typical thresholds used when collecting energy dispersive X-ray (EDX) spectra, particularly via the computer-controlled scanning electron microscopy approach (CCSEM-EDX), have been 1% or 2% for an element (N for the nitrogen in nitrate or S for the sulfur in sulfate). This depends on factors such as the substrate (i.e., is there a peak associated with it, such as with silicon, or not), spectral collection time, the accelerating voltage, and the sensitivity of the sample to the beam (i.e., does it degrade while being probed by the beam). The values are also different for N vs. S, as nitrogen is more difficult to detect via EDX. While we agree that this information would be very valuable, we cannot provide these estimates due to the semi-quantitative nature of EDX. This is especially true for nitrogen, which is at best semi-quantitative with EDX due to the low atomic number and peaks from carbon and oxygen. We have added further information regarding these thresholds in the methods section.

> Lines 232-237: "Commonly used thresholds for detecting an element with CCSEM-EDX are 0.5-1% (by mole fraction) (Hopkins et al., 2008), and depend on factors such as substrate, collection time of spectra, accelerating voltage, sensitivity of the sample to the electron beam, and the element being measured (i.e., EDX is less sensitive to N than S in this analysis). A rough rule of thumb is that when elements are not detected they are < 1% of particle mass."

Line 491-498: We found that during Asian dust events, aerosol Fe solubility at Qingdao (a coastal city of Northwest Pacific) was not elevated compared to that at desert regions (Chen et al., 2024). This suggested that Fe solubility was not significantly increased when Asian dust aerosol was transported to Qingdao, be consistent with what the authors suggested.

We have noted the similarity of our results for African dust to Asian dust and have added the suggested reference.

**References:**

Chen, Y. Z., Wang, Z. Y., Fang, Z. Y., Huang, C. P., Xu, H., Zhang, H. H., Zhang, T. Y., Wang, F., Luo, L., Shi, G. L., Wang, X. M., and Tang, M. J.: Dominant Contribution of Non-dust Primary Emissions and Secondary Processes to Dissolved Aerosol Iron, Environ. Sci. Technol., 58, 17355-17363, 2024.

Tang, M. J., Cziczo, D. J., and Grassian, V. H.: Interactions of Water with Mineral Dust Aerosol: Water Adsorption, Hygroscopicity, Cloud Condensation and Ice Nucleation, Chem. Rev., 116, 4205–4259, 2016.

Zhang, H. H., Li, R., Huang, C. P., Li, X. F., Dong, S. W., Wang, F., Li, T. T., Chen, Y. Z., Zhang, G. H., Ren, Y., Chen, Q. C., Huang, R. J., Chen, S. Y., Xue, T., Wang, X. M., and Tang, M. J.: Seasonal variation of aerosol iron solubility in coarse and fine particles at an inland city in northwestern China, Atmos. Chem. Phys., 23, 3543-3559, 2023.

REVIEWER #2

This research focused on studying the degree of dust mixing with sea salt and particulate pollutants. This includes several interesting techniques for the data analysis. Although the results are based on a rather short time measurements period, they are of interest. The topic is within the scope of Atmospheric Chemistry and Physics. This study may be published after addressing aa set of questions and comments listed below.

We thank the reviewer for their helpful comments.

Specific Comments:

C1. In the introduction the term 'aging' should be formally defined.

Line 60-61 authors said: The chemical change in the physiochemical properties of dust, herein referred to as "aging", can alter the water uptake properties of dust…

This is ambiguous and not a formal definition, the use of '…chemical change in the physiochemical properties…' sounds ambiguous.

Do the authors use the term aging as equivalent to internal mixing?

If so, the study concludes that dust particles (67% of them) are internally mixed with sea salt. Then, the title (…Lacks Indicators of Chemical Aging) is confusing.

We now clarify that chemical aging refers to changes in the physicochemical properties of dust due to heterogeneous and multiphase reactions. We also note that this is distinct from processes such as internal mixing of dust and sea salt due to coagulation.

> Lines 60-67: "The chemical change in the physiochemical properties of dust due to heterogeneous or multiphase reactions, herein referred to as "aging", can alter the water uptake properties of dust (Krueger et al., 2003; Laskin et al., 2005) thus affecting its light scattering efficiency (Bauer et al., 2007; Kandler et al., 2011; Levin et al., 1996), cloud droplet activation properties (Gibson et al., 2007; Kelly et al., 2007; Sullivan et al., 2009; Tang et al., 2016), and atmospheric lifetime (Abdelkader et al., 2015; Lance et al., 2013; Li et al., 2014; Wu et al., 2013). We note that chemical aging is distinct from other changes in the physicochemical properties of dust that may occur during transport such as coagulation."

C2. Line 31-32: Ion chromatography (IC) analysis of bulk nitrate, sulfate, and oxalate increase when African dust reaches Barbados, indicating dust aging.

How was this increase detected or measured? By comparing with other sites?

Please, explain in the text.

We now clarify that the increase in bulk nitrate, sulfate, and oxalate when African dust reaches Barbados is relative to background conditions at our site.

C3. Line 31-32.

English is not my mother's thong, but I think that the sentence should be rewritten.

What did increase? The analysis or the concentrations

Please, consider rewriting as:

Ion chromatography (IC) analysis indicates that the concentrations of bulk nitrate, sulfate, and oxalate increase when African dust reaches Barbados after transatlantic transport, indicating dust aging.

We thank the reviewer for this suggestion. Combined with C2, this sentence now reads,

> Lines 31-33: "Ion chromatography (IC) analysis indicates that the mass concentrations of bulk nitrate, sulfate, and oxalate increase, relative to background conditions, when African dust reaches Barbados after transatlantic transport, indicating dust aging."

C4. Line 80-81.

Authors cite Li-Jones & Prospero,(1998) describing that the sulphate mixed with dust may be linked to sulphur emissions in Europe.

More recent studies have linked the nss-sulphate mixed with Saharan dust to industrial emission in North Africa (see details in  https://doi.org/10.5194/acp-11-6663-2011 )

This updated information should be included.

We now cite Rodriguez et al, 2011 and note the findings from Izaña showing that North African emissions from industrial sources can also age dust.

> Lines 84-86: "In a more recent study from Izaña, African dust has been shown to acquire sulfate from industrial emissions from North Africa (Rodriguez et al., 2011)."

C5. Line 85-92. Authors cite several studies addressing dust aging.

Authors may be interested in studies on the changes of iron solubility during the westward trans-Atlantic transport of dust, e.g. https://doi.org/10.1016/j.atmosenv.2020.118092 , which includes the Barbados site.

We now cite the suggested paper.

> Lines 91-93: "In contrast, iron solubility was found to increase in Saharan dust during transatlantic transport, in part, due to chemical aging (Rodriguez et al, 2021)."

C6. Lines 103-112. Authors only cite biomass burning as a North African potential pollutant mixed with dust; the role of the North African industrial emissions is being omitted, and should be cited (see C4).

We now add reference to anthropogenic emissions from Northern Africa that can interact with dust. We note that we focused on biomass burning because smoke was found to be internally mixed with dust in our dataset.

> Lines 108-111: "In addition to anthropogenic emissions that mix with dust (Gaston et al., 2024; Rodriguez et al., 2011), the Sahelian burn season occurs during the boreal winter, often resulting in long-range co-transport of dust and smoke emissions that were specifically observed during the ATOMIC/EUREC$^4$A campaigns"

C7. Line 134-135. Authors cite NOAA's HYSPLIT model.

A reference is needed, e.g. https://doi.org/10.1175/BAMS-D-14-00110.1

 We have added the suggested reference.

C8. Section 3.3 Aerosol Mixing State Analysis, actually defines a sampling system rather than an analysis system. Shouldn't his section be called something as Section 3.3 Aerosol sampling for mixing state analysis??

How are the samples collected with this system analyzed and what is actually being determined?

This should be included.

 We have changed the title of this section to read, "*Aerosol Collection for Single Particle Mixing State Analysis*"

We also applied the reviewer's comment to our other sections of the paper to clarify the technique used and the aerosol physicochemical property analyzed.

C9. Line 301. What particle type is smoke? What type of analytical technique was used to measure it ?,.. is it elemental carbon?

We now define smoke particle identification with EDX when other particle types are introduced.

> Lines 242-244: "Smoke particles were identified as containing a combination of C, O, S, and K, as shown previously at this site (Royer et al. 2023) and other sites (Bondy et al. 2018; Olson et al. 2019)."

We now clarify that CCSEM/EDX was used to determine the presence of smoke from African wildfires.

Lines 318-322: "Results from CCSEM/EDX analysis revealed the presence of both marine particles such as sea spray, aged sea spray, organics, and sulfates, as well as continental particle types including dust, internally mixed dust and sea salt, internally mixed dust and smoke from African wildfires, and externally mixed smoke from African wildfires that are described in detail in Royer et al., 2023."

We also clarify in the Methods that SEM provides particle images and morphology while EDX provides characteristic elemental spectra of each particle

Lines 212-214: "SEM provides images, size, and morphology of individual particles while EDX measures the semi-quantitative elemental composition of each particle."

C10. Line 310 and in other sections of results, it would be helpful to remind the size ranges of supermicron, 1-5 mm ?, not all > 1 mm

The reviewer brings up an important point that the upper bound of particles sampled for single particle and bulk chemical analysis is different. For the filter samples described in Section 4.1, we now clarify that the bulk aerosol analysis is for our filter samples while the supermicron particle size cut is 5 μm for particles analyzed by CCSEM/EDX. We clarify this point in Section 4.2 and throughout the paper now.

Lines 329-335: "While dust mass concentrations from our filters for the submicron (<1.3 μm) and supermicron (>1.3 μm) aerosol loading are similar at 14.38 μg/m$^3$ and 12.28 μg/m$^3$, respectively, similar to results found for Asian dust (Zhang et al., 2023), CCSEM analysis reveals that dust only makes up 21% and 4% of the submicron (<1 μm) and supermicron (1-5 μm) aerosol loading by particle number, respectively. We note that part of this discrepancy could be because the CCSEM analysis is restricted to an upper limit of 5 μm in diameter."

We also clarify in our section titles which technique was used and what the property measured was.

C11. Lines 312-314: ... large difference in the number fractions of submicron and supermicron dust particles, where dust only makes up 21% and 4% of the submicron and supermicron aerosol loading by particle number, respectively.

This result is actually something expectable, not surprising, for any type of aerosol, not only for dust. Aerosol particles are more abundant in number concentration in the submicron than in the supermicron fraction, even of the supermicron fraction has a much large dust mass. Several studies are showing that the number size distribution is shifted to small particle size, whereas mass size distribution is shifted to large/coarse particle size. For this reason, in my opinion, it is not justified to say what the authors said in lines 314-316: 'This

discrepancy is not only due to the difference in individual particle mass between sub- and supermicron particles but also due to the oversimplification of dust particles in bulk analysis'. What does the author mean when saying `oversimplification of dust particles in bulk analysis´.

In my modest opinion, lines 314-316 could be deleted, such oversimplification is not true.

We agree with the reviewer. We have reworded this section.

Lines 328-335: "In Figure 2a, particle types are plotted as number fractions of the aerosol loading as a function of aerosol diameter. While dust mass concentrations from our filters for the submicron (<1.3 μm) and supermicron (>1.3 μm) aerosol loading are similar at 14.38 μg/m$^3$ and 12.28 μg/m$^3$, respectively, similar to results found for Asian dust (Zhang et al., 2023), CCSEM analysis reveals that dust only makes up 21% and 4% of the submicron (<1 μm) and supermicron (1-5 μm) aerosol loading by particle number, respectively. We note that part of this discrepancy could be because the CCSEM analysis is restricted to an upper limit of 5 μm in diameter."

C12. Fig. 2 and text lines 317-373. Authors use N to refer to nitrogen and S for sulphur, this is strictly OK, however, N is usually used to refer the number of particles or number concentration (N) and S the particle surface (S), so, it is actually confusing for the reader, in fact I got confused. I suggest to the author describe in the figure caption that N and S reefer to nitrogen and sulphur, respectively. Even in the text (317-341) it would also be useful to state clearly that N and S actually mean. This is just a suggestion that will make the text more friendly to read.

We thank the reviewer for this helpful suggestion and have amended the text as suggested in this section, in the figure captions and throughout the manuscript.

C13. Lines 343-373. Can the author segregate the presence of natural calcium sulphate, i.e. gypsum or anhydrite? . This should, at least, be cited in the text. The same of natural sea-salt sulphate, which is cited in the text.

From our CCSEM/EDX results, we cannot infer mineralogy. However, we can infer if the sulfur present indicates heterogeneous processing, particularly for sea salt particles where the loss of chloride (Cl) indicates chemical aging with a compound such as sulfuric acid. For dust particles, we took a conservative approach and assume any sulfur present on the dust is a product from chemical aging.

C14. Lines 368-373. Sulphate mixed with dust includes a portion of ammonium sulphate and other sulphate salts, as calcium sulphate linked to natural gypsum and dust coating

with sulphate linked to industrial emissions in central Sahara, e.g. see the oil refineries located in Algeria (Ourgla, Hassi Mesahoud), fig. 2, fig 10 described in https://doi.org/10.5194/acp-11-6663-2011

We now cite Rodriguez et al 2011 for the observation of submicron ammonium sulfate. As indicated in our results, we did not observe much sulfate on transported dust particles.

C15. Section 4.4 TOF-SIMS Imaging. This section may be summarized.

We have changed the title of this section to read "*TOF-SIMS Imaging of Chemical Markers of Aerosol Aging*"

C16. Line 446. I would not say that traditional methods oversimplify, I actually think that the methodology used by the authors and traditional bulk analysis are complementary. In fact, traditional bulk analysis allows a better quantification and provides a global picture of the aerosol composition load, that may be complemented by the analysis presented by authors in this study.

We agree with the reviewer and have removed the word "oversimplified".

Lines 469-474: "Traditional methods for studying dust aging often measure dust concentrations (or their proxies) and soluble materials extracted from aerosol filters have historically used correlations between nss-sulfate, oxalate, and nitrate and dust mass concentrations to prove the presence of dust aging. However, these methods have been unable to determine the mixing state of dust and, thus, whether dust (or another aerosol type) is actually undergoing chemical aging (Chen & Siefert, 2004; Li-Jones & Prospero, 1998)."

C17. Lines 496-498. Dust may be an important source of nutrients, even if not aged. For example, dust provides iron to the ocean. Iron contained in dust has a low solubility (0.5% aprox), it may increase to 5-7% after several days in the air. Even if iron – dust is not processed these inputs of insoluble iron are important since some marine species may process such dust particle contains insoluble iron, this is the case of trichodesmium, who is able to digest insoluble iron – dust by using the trichomes ( https://doi.org/10.1038/s41396-019-0505x , https://doi.org/10.1016/j.isci.2021.103587 ). In fact, the migration of some tropical tuna tracks the seasonal shift of dust inputs to the Atlantic, even of dust is not aged (see details in https://doi.org/10.1016/j.atmosenv.2023.120022 ). In summary, to say (496-498) 'The lack of aging on mineral dust observed in this study suggests African dust may not be as important of a source of bioavailable nutrients for the North Atlantic as previously assumed' is too vague and general; in my modest opinion such sentence should be avoided.

**Citation**: https://doi.org/10.5194/egusphere-2024-3288-RC2

We agree with the reviewer's point here that other factors beyond chemical aging impact nutrient solubility. We have reworded our sentence.

> Lines 522-525: "The lack of chemical aging on mineral dust observed in this study suggests that the chemical aging of dust plays a limited role in observed increases in nutrient solubility during transatlantic transport, consistent with findings for Asian dust (Chen et al, 2024)."

REFERENCES CITED

Gaston, C.J., Prospero, J.M., Foley, K., Pye, H.O.T., Custals, L., Blades, E., Sealy, P., & Christie, J.A. (2024) Diverging trends in aerosol sulfate and nitrate measured in the remote North Atlantic on Barbados are attricuted to clean air policies, African smoke, and anthropogenic emissions. *Atmospheric Chemistry & Physics*, *24*(13), 8049-8066.

Olson, N.E., May, N.W., Kirpes, R.M., Watson, A.E., Hajny, K.D., Slade, J.H., Shepson, P.B., Stirm, B.H., Pratt, K.A., & Ault, A.P. (2019) Lake spray aerosol incorporated into Great Lakes Clouds. *ACS Earth & Space Chemistry*, *3*, (12), 2765-2774.

Rodriguez, S., Alastuey, A., Alonso-Pérez, S., Querol, X., Cuevas, E., Abreu-Afonso, J., Viana, M., Pérez, N., Pandolfi, M., & de la Rosa, J. (2011). Transport of desert dust mixed with North African industrial pollutants in the subtropical Saharan Air Layer. *Atmospheric Chemistry & Physics*, *11*, 6663-6685.

Rodríguez, S., Prospero, J.M., López-Darias, J., García-Alvarez, M.-I., Zuidema, P., Nava, S., Lucarelli, F., Gaston, C.J., Galindo, L., & Sosa, E. (2021). Tracking the changes of iron solubility and air pollutants traces as African dust transits the Atlantic in the Saharan dust outbreaks. *Atmospheric Environment*, *246*, https://doi.org/10.1016/j.atmosenv.2020.118092.

Stein, A.F., Draxler, R.R., Rolph, G.D., Stunder, B.J.B., Cohen, M.D., & Ngan, F. (2015). NOAA's HYSPLIT atmospheric transport and dispersion modeling system. *Bulletin of the American Meteorological Society*, *96*(12), 2059-2077.

van Herpen, M.M.J.W., Li, Q., Saiz-Lopez, A., Liisberg, J.B., Röckmann, T., Cuevas, C.A., Fernandez, R.P., Mak, J.E., Mahowald, N.M., Hess, P., Meidan, D., Stuut, J.-B.W., & Johnson, M.S. (2023). Photocatalytic chlorine atom production on mineral dust-sea spray aerosols over the North Atlantic. *Proceedings of the National Academy of Sciences*, *120*(31), https://doi.org/10.1073/pnas.2303974120.